# Investigation of Polyol Process for the Synthesis of Highly Pure BiFeO_3_ Ovoid-Like Shape Nanostructured Powders

**DOI:** 10.3390/nano10010026

**Published:** 2019-12-20

**Authors:** Manel Missaoui, Sandrine Coste, Maud Barré, Anthony Rousseau, Yaovi Gagou, Mohamed Ellouze, Nirina Randrianantoandro

**Affiliations:** 1Institut des Molécules et Matériaux du Mans (IMMM)–UMR CNRS 6283, Le Mans Université, Avenue Olivier Messiaen, 72085 Le Mans Cedex 9, France; manel.missaoui.11@gmail.com (M.M.); Maud.barre@univ-lemans.fr (M.B.); anthony_rousseau@univ-lemans.fr (A.R.); Nirina.Randrianantoandro@univ-lemans.fr (N.R.); 2Laboratoire des Matériaux Multifonctionnels et Applications (LMMA)—LR16ES18, Faculté des Sciences de Sfax (FSS), Route de Soukra km 3.5, B.P.1171, 3000 Sfax, Tunisie; mohamed.ellouze@fss.rnu.tn; 3Laboratoire de Physique de la matière condensée (LPMC), Université de Picardie Jules Verne, Chemin du Thil, CS 52501 80025 Amiens Cedex 1, France; yaovi.gagou@u-picardie.fr

**Keywords:** bismuth ferrite, polyol process, nanoparticles, multiferroic, cycloid

## Abstract

Exclusive and unprecedented interest was accorded in this paper to the synthesis of BiFeO_3_ nanopowders by the polyol process. The synthesis protocol was explored and adjusted to control the purity and the grain size of the final product. The optimum parameters were carefully established and an average crystallite size of about 40 nm was obtained. XRD and Mössbauer measurements proved the high purity of the synthesized nanostructurated powders and confirmed the persistence of the rhombohedral R3c symmetry. The first studies on the magnetic properties show a noticeable widening of the hysteresis loop despite the remaining cycloidal magnetic structure, promoting the enhancement of the ferromagnetic order and consequently the magnetoelectric coupling compared to micrometric size powders.

## 1. Introduction

Multiferroic materials present a combination of electrical, magnetic and/or elastic features. They were classified in two types: type (I), in which the ferroelectricity coexists with the magnetic order spontaneously, and type (II), where the ferroelectricity is induced by the magnetic order (e.g., TbMnO3). The prototype material bismuth ferrite (BiFeO_3_), denoted as BFO, belongs to the type (I). Actually, the ferroelectric order in BFO was described by the lone pair model [1] of bismuth (Bi, 6s^2^) which causes, by the hybridization with oxygen orbitals, a displacement of bismuth from the center of the perovskite. Furthermore, early studies have proven the existence of G-type antiferromagnetic order and weak ferromagnetic order, and that was explained by the Dzyaloshinskii-Moriya (DM) spin canting interaction [2]. However, in order to allow the unusual coexistence of both ferroelectric and magnetic orders, a structural distortion of the crystal cell and of the rotating FeO_6_ octahedra is required.

From practical point of view, BFO is the unique simple perovskite that exhibits spontaneous magnetoelectric coupling at room temperature, endowed with a high structural, magnetic and ferroelectric stability in a large range of temperatures. Indeed, the ferroic transitions occur at TN (bulk) ∼643 K and TC (bulk) ∼1103 K, depending on the quality of the sample [3]. However, two major obstacles are inhibiting the understanding of intrinsic BFO properties and the use of such a rare multiferroic and multifunctional material. First of all, the synthesis of the pure BFO phase is a hard task to accomplish, due to the thermodynamic competition between the BFO phase and the Fe-rich and Bi-rich phases (known as millite and sillenite, respectively) [4,5]. Second of all, beneath the magnetic transition temperature, BFO presents a complex incommensurate magnetic structure, repeating each 62 nm, causing the compensation of most of the resulting magnetic moments which weakens the ferromagnetism and lowers the magnetoelectric coupling. Therefore, the efficiency of BFO, particularly for ambient applications, can be limited. By stabilizing the BFO phase and suppressing the cycloidal structure, this material can be used in a variety of technological devices, such as spintronic devices, Read-Write units for magnetic devices [6], gas sensors [7], capacitors, Magnetic Resonance Imaging (MRI) and in the biomedical field [8], thanks to its magnetoelectric coupling [9] and nonlinear optical properties, and also in photovoltaic and photocatalysis applications [10,11,12].

For these reasons, many chemical routes were developed to synthesize the pure BFO phase and also to control the grain size and morphology, and it was found that in most cases, BFO nanoparticles exhibit much better chemical and physical properties compared to micrometric powders [8,13,14].

Among the numerous soft chemical routes which were published, no study has been reported up to date about the synthesis of BFO by the polyol process, despite the popularity of this technique basically because of its simplicity, versatility and the possibility to obtain reproducible [15] high quality materials with low-cost starting products and simple equipment [16,17].

Initially, the polyol process was used to produce metals [18] and alloys [19] and hence noble metals (Au, Pt…) due to its reducing capacity, and then it was enlarged for the synthesis of oxides [20] and perovskites [21] displaying improved physical properties, and most of the time totally new emerging electrical and optical features as a consequence of the changed shape, color, and particle size and structure [22] and the enlarged specific area [23,24]. It is interesting to recognize the large range of the possible sizes, going from the mesoscale down to 4 nm [25], which can be useful if one needs to investigate the overturning of physico-chemical properties of some phases based on the particles size. Actually, the polyol process gained its high reliability from its great efficiency to inhibit the agglomeration of particles, to provide a remarkable uniformity of grain shape and narrow size distribution, and to be able to mix reactants at the molecular level [26]. Those outstanding characteristics actually refer to the role of polyols which act as solvents and reducing agents and stabilize the nucleation and growth of particles by limiting their agglomeration, due to their chelating effect. They also help to lower the synthesis temperature [27], since a major part of the nucleation and interdiffusion has already happened inside polyol medium during the reflux; in some cases, final phases can be obtained without further heat treatment [26]. Also, the colloidal suspension of the mixed elements in the dielectric polyol medium enables the growth of specific grain morphologies (nanopeanuts [28], nanowires [29], nanocubes [30], flower-like shapes [31]). However, despite the overall easiness of manipulating polyol synthesis, the external conditions must be vigorously controlled to ensure the reproducibility of the obtained phase and the maintaining of the morphology of the nanoparticles. In our experiment, we were confronted with a major difficulty related to high room temperature and humidity in the hot season that affected the quality of the starting nitrate salts we used and resulted in an impure compound.

In association to our research purpose, many studies reported a surprising high magnetization of iron [32] and ferrite alloys, e.g., FeCo [33,34,35,36] and NiFe [37], once synthesized by the polyol process, with a remarkable control of their size and form. With this aim, we explored the polyol process for the synthesis of BFO nanoparticles. The optimization of the heat treatment of the precursor obtained by the polyol process and the effect of the nanostructuration of the powder on the magnetic properties, in comparison with micrometric size powders, are the subject of another article [38]. In this work, we focus on the effect of different synthesis parameters on the structure and microstructure of the powders and on their impact on the magnetic response.

## 2. Materials and Methods

### 2.1. Synthesis of the BiFeO_3_ Nanoparticles

Bismuth nitrate (Bi(NO_3_)_3_.5H_2_O) and iron nitrate (Fe(NO_3_)_3_.9H_2_O) (Alfa Aesar, Thermo Fisher, Karlsruhe, Germany, 98%) were dissolved in diethylene glycol (DEG, Alfa Aesar, Thermo Fisher, Karlsruhe, Germany, 99%), with a 1:1 molar ratio, and heated up to 200 °C under reflux for 2, 3 or 6 h. During heating, the orange transparent solution started to darken on reaching 150 °C and a precipitate started to appear. The dissolution of the nitrates was done in two different ways: by direct magnetic stirring in DEG while heating, or by sonication until the total dissolution of the salts before the reflux heating step. The obtained brown precipitates were washed with acetone and centrifuged three times to eliminate DEG, as much as possible. Then they were collected in a beaker with additional acetone, sonicated for 15 min to disperse the precipitate (“for quicker dryness”) and then dried at 120 °C.

The resulting precursors were ground in agate mortar and disposed in a platinum crucible to be heated, under air, at 500 °C for 2 h with a heating rate of 5 °C/min and then quenched at room temperature in order to remove the organic groups from the precursor and then to allow the crystallization of the BFO compound. The organic materials were mainly of the hydroxyl, carbonyl, alkyl or nitro groups. It has to be pointed out that, when the precursor is placed in the crucible before heating, it must be well spread in a way to form only a thin layer. Otherwise, impurities form during the thermal treatment [38].

### 2.2. Characterization Techniques

The purity of the powders was checked by powder X-ray diffraction (XRD). The crystallite size was also determined according to the Scherrer equation for the Bragg reflection (0 1 2)_hex_ (the contribution of the apparatus on the FWHM being deduced). The patterns were recorded at room temperature on a PANalytical θ-θ Bragg Brentano X’pert MPD PRO diffractometer (Malvern, Panalytical B.V., Almelo, Netherlands) equipped with a Cobalt source (λ_KαCo_ = 1789 Å).

The morphology and grain size of nanopowders were investigated with a JEOL JEM 2100 HR transmission electron microscope (JEOL Ltd., Tokyo, Japan) (TEM).

The magnetic behavior of BFO powders was explored by Mössbauer spectrometry, employing a ^57^Co source, in transmission configuration with a constant velocity of 12 mm·s^−1^.

Magnetic results were obtained using Quantum Design PPMS DynaCool (Quantum Design, North America, San Diego, CA, USA) which offers versatile vibrating sample magnetometer (VSM), AC and DC susceptibility measurements. This equipment offers a continuous low or high temperature control sweep mode and a precise magnetic field, integrating a cryopump compatible with all the other measurement options from 0 up to 9 T magnetic field and a temperature range from 2 to 1000 K.

## 3. Results and Discussion

### 3.1. Effect of Synthesis Parameters on the Microstructure

#### 3.1.1. Effect of the Solvent

Diethylene glycol shows more stability and can be considered as a better candidate for reflux at different temperatures below its boiling temperature of 246 °C. Thus, it was proven to be useful for the synthesis of nanosized perovskite materials [21,24]. However, several reports have shown that the use of ethylene glycol for the synthesis of various types of compounds [39,40] has an impact on the microstructure [41].

In order to determine if the use of ethylene glycol (EG) instead of diethylene glycol (DEG) has an impact on the structure and microstructure of the powders, a synthesis was made with EG (Alfa Aesar, Thermo Fisher, Karlsruhe, Germany, 99%) as solvent instead of DEG. As the boiling temperature of EG is 189 °C, the heating under reflux of the solution was realized at 190 °C. In both polyols, the dissolution of the iron nitrate was easy, the solution, initially transparent, becoming orange. Indeed, simple stirring was enough for its complete dissolution. This may be due to the dissolving role of polyols on transition metals [16]. However, the bismuth nitrate cannot be dissolved at room temperature with magnetic stirring. Actually, bismuth grains remain visible until the solution starts to become obscure when the temperature reaches 150 °C during the heating under reflux. After centrifuging and drying the precipitate, the precursors were heat treated at 500 °C for 2 h under air. As evidenced by XRD (Figure 1), the use of EG leads to the development of secondary phases, while the DEG was suitable for the formation of pure BFO (rhombohedral R3c symmetry) by the polyol process.

#### 3.1.2. Effect of the Reflux Temperature and Duration

In order to determine if the reflux conditions have an influence on the crystallite size and the purity of the powder, the temperature and duration of the heating of the solution under reflux were varied (C = 0.06 mol·L^−1^). On one hand, in addition to the reflux temperature of 200 °C that was initially used, two other reflux temperatures were tested: 190 °C and 230 °C (reflux duration of 3 h). On the other hand, the initial solution containing bismuth and iron nitrates dissolved in DEG was heated under reflux for durations of 2, 3 or 6 h, the reflux temperature being fixed at 200 °C.

In the case of various heating temperatures of the solution under reflux, the purity of the final powders was largely affected (Figure 2). For the synthesis performed at 230 °C, in order to be closer to the DEG boiling temperature, the obtained precursor was already crystallized and this crystallization during reflux would not allow the formation of pure BFO powder, as it was always observed when a problem occurred during the synthesis of other precursors which were already crystallized. Heating under reflux at 190 °C led, after the heat treatment of the precursor at 500 °C for 2 h, to the presence of about 10 wt% of Bi_2_O_3_ (or Bi-rich phase, that present quite the same XRD pattern) in addition to the BFO phase.

It can be also noted that any unexpected stop of heating, or any bad regulation of temperature during the heating under reflux step, exceeding 200 °C ± 5 °C, led to the perturbation of the nucleation and/or growing process, as the solution rapidly cooled down, and then led to an impure final powder after heat treatment at 500 °C under air.

Concerning the effect of the duration of the heating under reflux, the powders obtained after heat treatment of the precursor at 500 °C, under air, were pure according to XRD patterns, whatever the duration (2, 3 or 6 h) (Figure 3b). However, Mössbauer spectra revealed the presence of 7% of paramagnetic impurity (here Millite) when the solution was refluxed for only 2 h, as shown in Figure 3a. For reflux durations of 3 or 6 h, no impurity was observed by Mössbauer spectrometry, which is more sensitive than XRD for compounds containing iron. Moreover, STEM (Scanning Transmission Electron Microscopy) image and X maps were performed for the sample synthesized with a reflux of 3 h at 200 °C and heat treated at 500 °C for 2 h under air (Figure 4). They evidence that the particles were homogeneous in composition. It can also be noted that the grains shape slightly tended towards an ovoid shape for the longer refluxes (6 h), as observed by TEM (Figure 9a–d).

It indicates that a reflux time of at least 3 h is probably necessary to let the precipitation of iron and bismuth proceed. Eventually, this longer time also allows a partial dissolution of the obtained precipitate and again precipitation of the dissolved species. Consequently, the final precipitate presents a higher homogeneity in the distribution of the different metals and leads to pure BFO after the thermal treatment.

#### 3.1.3. Effect of the Dissolution Method

As explained previously, iron nitrate can be instantly dissolved in DEG, contrary to bismuth nitrate, which is insoluble in polyols at room temperature, under stirring. Thus, in order to ensure the equal dispersion of both reagents in the solution before starting the nucleation process under reflux-heating, the suspension was sonicated in a beaker for about 20 min until the complete dissolution of bismuth nitrate. Then, the homogenous orange transparent solution was poured into the reactor and the rest of the liquid was collected with 2 mL of ethanol.

The purity of both powders issued from the dissolution of the reagents by sonication or only by magnetic stirring (without any prior sonication) was determined by XRD and Mössbauer measurements for the concentration [Bi + Fe] of 0.06 mol·L^−1^ (Figure 5). Whatever the dissolution method employed, after a heat treatment of precursors at 500 °C under air, the resulting powder is pure (only BFO phase) according to the XRD characterizations. This result is confirmed by the analysis of the Mössbauer spectra of the different samples, as there is no variation of the hyperfine parameter values of both samples, as shown in Figure 5b. Moreover, the sonication does not drastically influence the crystallite size as it is of around 42 nm, with or without sonication.

#### 3.1.4. Effect of the Concentration

In order to reduce the BFO particle size, we varied the molar concentration of cations [Bi + Fe] in the solutions as follows: 0.1, 0.06, 0.04, 0.02 mol·L^−1^, which are considered to be very low for chemical soft methods. The precursors obtained after the heating of the solution under reflux for 3 h, were heat treated at 500 °C for 2 h. The final powders were pure for the concentrations 0.06 and 0.04 mol·L^−1^ (Figure 6), while a higher concentration of 0.1 mol L^−1^ led to a crystallized precursor, the powder obtained after the heat treatment at 500 °C for 2 h under air always containing, with such crystallized precursors, impurities in addition to the BFO phase. For the lower concentration C = 0.02 mol·L^−1^, the final powders contained impurities (8 wt% of Bi_2_O_3_). Moreover, the formation of the precipitate during the heating of the solution under reflux was reduced and in the case of the concentration 0.02 mol·L^−1^, the quantity of precipitate that was collected was very low.

These results evidence that the concentration range that allow us to obtain pure BFO powders is reduced, the crystallite size being of about 43 nm for both C = 0.04 or C = 0.06 mol·L^−1^.

#### 3.1.5. Effect of the Addition of Hydroxide or Hydronium Ions

In order to control the morphology of the powders, the influence of the addition of hydroxide or hydronium ions was determined. Indeed, the addition of OH^−^ ions can, for instance, influence the morphology of particles [42]. More generally, the presence of hydronium or hydroxide groups can influence particle growth and stability.

In this way, different acids (nitric acid, citric acid or acetic acid) and/or urea (that leads to the formation of hydroxide groups when it decomposes under heating of the solution) were added to the optimized solution (bismuth and iron nitrates dissolved in DEG with C = 0.06 mol·L^−1^). This solution was then heated under reflux at 200 °C for 3 h and the precursor was heat treated at 500 °C for 2 h under air.

Pure BFO powders were obtained, as revealed by the XRD patterns (Figure 7), when only urea or both urea and acetic acid were added to the mixture of DEG and nitrate salts. This could be due to the fact that the addition of acetic acid enhanced the dissolution of bismuth nitrate by magnetic stirring before the heating of the solution under reflux. However, TEM observations (Figure 9g,h) showed more agglomerated grains and an undefined morphology, the crystallite size determined from the XRD patterns being approximately the same than without these additions (Table 1). Moreover, the addition of concentrated nitric acid led to the formation of a bismuth-rich phase, as well as citric acid or only acetic acid. The obtained phases are summarized in Table 1 and the crystallite sizes are indicated.

#### 3.1.6. Effect of the Addition of Water or of a Surfactant

In order to control the microstructure of the powders, the addition of water or of a surfactant, polyvinylpyrrolidone (PVP), was also tested.

Deionized water (h = [water]/[Bi + Fe] = 10) was added to the reference solution with a concentration of 0.06 mol·L^−1^. By following the same process that we established previously (heating of the solution under reflux for about 3 h at 200 °C and heat treatment of the precursor at 500 °C for 2 h under air), we obtained a BFO phase with a particle size of 48 nm but with the presence of 6 wt% of Bi_2_O_3_ (Figure 8). Moreover, TEM observation evidenced that water enhanced grain agglomeration but without any difference in morphology of the grains (Figure 9e).

For the surfactant, PVP, the first experiment done with 1.44 g in 15 mL of DEG did not lead to the formation of a precipitate of Bi and Fe, and the final solution remained orange transparent with agglomerated PVP that do not completely dissolved. After the optimization of the required quantity to ensure the occurrence of the precipitation, we determined that the value of C = 4.10^−4^·mol·L^−1^ of PVP leads to a higher purity of the powder. However, 9 wt% of the Bi_2_O_3_ impurity was detected (Figure 8). Moreover, the crystallite sizes obtained with PVP polyol solution or without, are quite similar (39 nm with PVP instead of about 43 nm). In addition, TEM images do not evidence a lowering of the agglomeration of the powders, as it could be expected (Figure 9h).

#### 3.1.7. Conclusion on the Effect of the Different Parameters

The effects of the different parameters on the purity of the powders and on the microstructure (Figure 9) were determined. The best microstructures, with a low grain size and a higher purity, were obtained for the simplest synthesis, based on a nitrate mixture in DEG and heating under reflux of the solution of at least 3 h. Ovoid-like shape grains can be well defined and the grains seem to be less agglomerated (Figure 9a–c) compared to the other preparations, according to all the images observed for each sample. The specific ovoid shape was not achieved for short reflux (2 h in Figure 9d), and the grains tended to agglomerate more when supplementary agents were added (see in Figure 9e–h referring respectively to the addition of water, urea, acetic acid and urea and finally PVP). Therefore, we selected the best protocol that allows homogenous, well-formed grains of narrowly distributed sizes that can be obtained in a reasonable amount of time, which corresponds to 3 h of heating of the solution of nitrates salts in DEG under reflux at 200 °C, and then a heat treatment of the precursor at 500 °C for 2 h under air.

### 3.2. Magnetic Properties

In this section, we present the physical properties of synthesized BFO nanostructured particles obtained from the best protocol, with C = 0.06 mol·L^−1^, 3 h of reflux of nitrate salts in DEG at 200 °C, the precursor being heat treated at 500 °C for 2 h under air.

The microstructural and structural studies reported above showed that the final product consists of a pure BFO phase. XRD and Mössbauer experiences have not detected the presence of secondary phases. Figure 10 shows that the Mössbauer spectra, recorded at room and liquid nitrogen (77 K) temperatures, consist of a pure BFO phase, recognizable by asymmetric lines.

Mössbauer data were analyzed using MOSFIT, a Lorentzian line fitting program [43] allowing for the use of a discrete distribution of hyperfine parameters. Based on the cycloidal model described above, we obtained an Isomer Shift of δ = 0.41 (0.5) mm·s^−1^, a quadripolar splitting of 2ε = 0.46 (0.51) mm·s^−1^, and a hyperfine field of B_hf_ = 49.3 (54.5) T at room temperature (nitrogen liquid). These values are typical of the high-spin Fe^3+^ and clearly show that synthesized BFO particles have a Fe^3+^ valence state only. From these Mössbauer data it is worth noting that, even though the crystallite size (of around 40 nm) is lower than 62 nm, the obtained BFO nanoparticles conserve the cycloidal spin structure.

M(H) loop measurements at different temperatures, ranging from 4 K to 300 K, were carried out to check the macroscopic behavior of the magnetization of the BFO nanostructured particles. Figure 11 shows clearly that the M(H) curve for the BFO nanoparticles (≈40 nm) made with the polyol process exhibits a non-saturated ferromagnetic behavior compared to a micrometric size BFO powder (≈200 nm) with a G-type antiferromagnetic order.

The appearance of the hysteresis loop indicates that when the crystallite size of BFO becomes lower than the period of its cycloidal spin structure, the compensation of magnetic moments in the G-type antiferromagnetic order is no longer achieved and a ferromagnetic-type order appears. This phenomenon is the consequence of the break of symmetry of the incommensurable cycloidal spin structure due to the finite size of particles. In other words, if the polyol method allows a size reduction below the critical size value of 62 nm, the nanoparticles obtained preserve the non-collinear magnetic order, as observed from Mössbauer spectrometry, but have a larger spontaneous magnetization. The measured values of coercive field and remanence are in order of µ_0_HC ≈ 0.6 T and M_r_ ≈ 0.07 Am^2^ kg^−1^, respectively.

## 4. Conclusions

Pure ovoid-like shaped BFO nanostructured particles were successfully synthesized for the first time by the polyol process using DEG as a solvent. The synthesis parameters were controlled and allowed us to obtain an average particle size of ∼40 nm, which is under the periodicity of the modulated cycloidal arrangement of iron spins. Surprisingly, the asymmetric Mössbauer spectrum revealed the persistence of the cycloidal magnetic structure at this scale. However, a spectacular room temperature hysteresis loop was registered, which explains the enhancement of the ferromagnetic order compared to the antiferromagnetic (AFM) bulk material. This exaltation of the macroscopic magnetization can be explained by a symmetry breaking, due to the finite size of the BFO nanostructured particles which induces a non-compensation of the total magnetization of Fe^3+^ sublattices in the cycloidal spin structure.

To summarize, the physical characterization presented in this paper showed a transformation of the overall magnetic order and an exalted spontaneous magnetization when the particle size was reduced below 62 nm. This can improve the magnetoelectric coupling and makes these BFO nanostructured powders low-cost potential candidates for spintronic technology applications in various domains, such as a magnetic and electric field sensors, MRA memories and electromagnetic devices.

## Figures and Tables

**Figure 1 nanomaterials-10-00026-f001:**
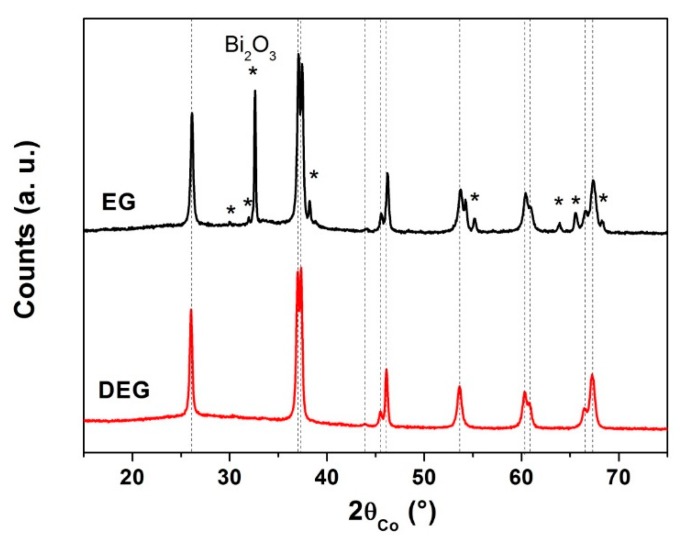
XRD patterns showing the effect of the nature of the polyol (ethylene glycol (EG) or diethylene glycol (DEG)) on the purity of bismuth ferrite (BFO) powders (precursors heat treated at 500 °C for 2 h). (Dotted lines correspond to the BFO phase).

**Figure 2 nanomaterials-10-00026-f002:**
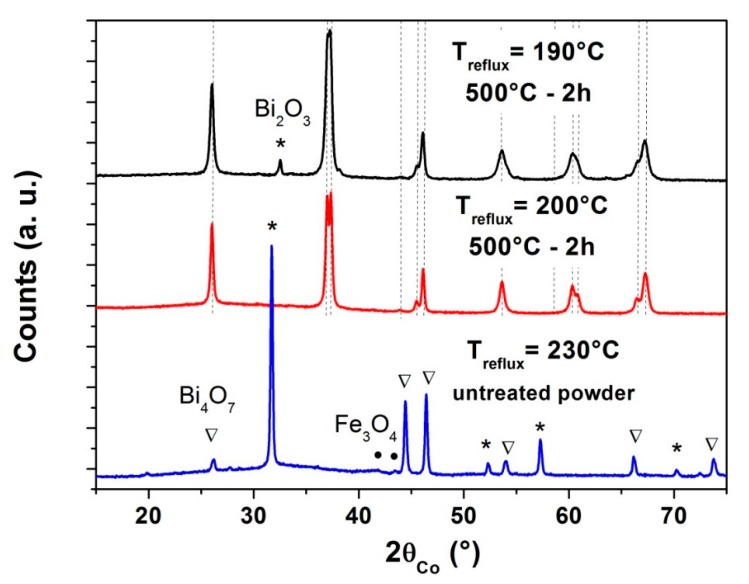
XRD patterns for the determination of the suitable heating temperature under reflux with DEG (C = 0.06 mol·L^−1^) in order to obtain pure final powder after a heat treatment at 500 °C (The crystallized precipitate obtained at 230 °C was not heat treated at 500 °C) (Dotted lines correspond to the BFO phase).

**Figure 3 nanomaterials-10-00026-f003:**
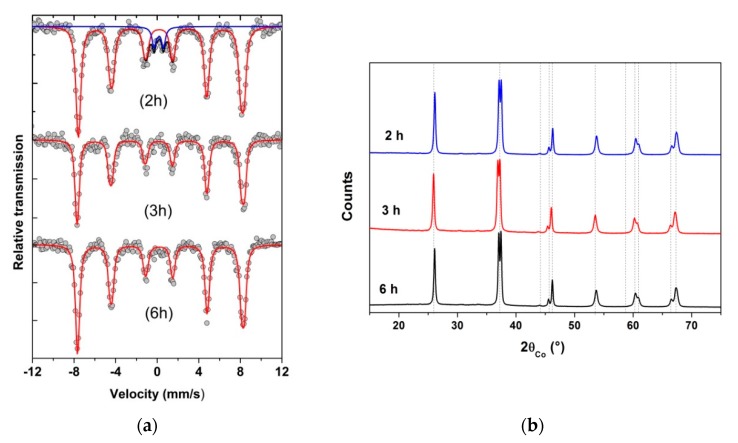
(**a**) Mössbauer spectra, registered at room temperature, showing the presence of a 7% paramagnetic impurity (Millite phase) for the shorter reflux duration 2 h (C = 0.06 mol·L^−1^) while in the (**b**) XRD pattern of the same samples, only a BFO phase is detected for 2, 3 or 6 h of reflux.

**Figure 4 nanomaterials-10-00026-f004:**
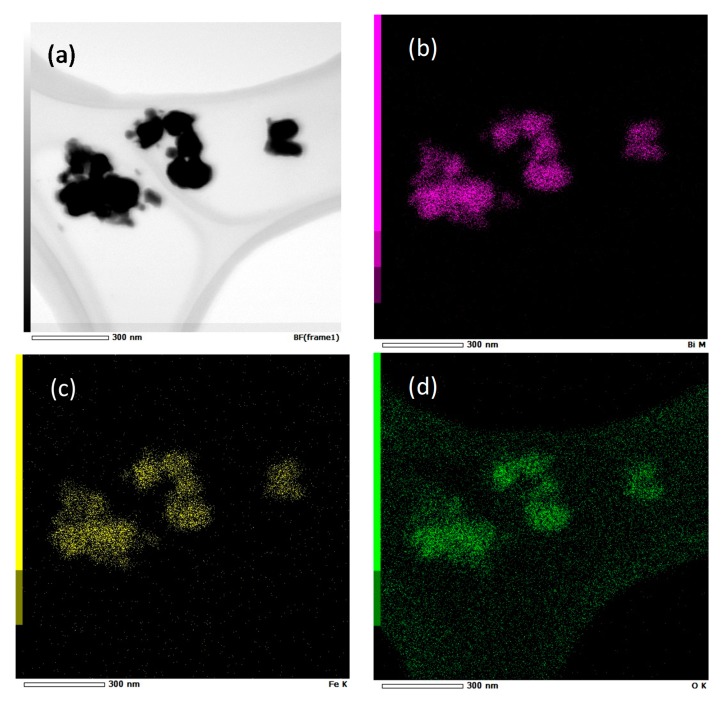
STEM image (**a**) and X maps of the precursor prepared with C = 0.06 mol·L^−1^ and a reflux of 3 h at 200 °C and then heat treated at 500 °C for 2 h, evidencing the homogeneity in composition of the particles. (pink: bismuth (**b**), yellow: iron (**c**), green: oxygen (**d**)).

**Figure 5 nanomaterials-10-00026-f005:**
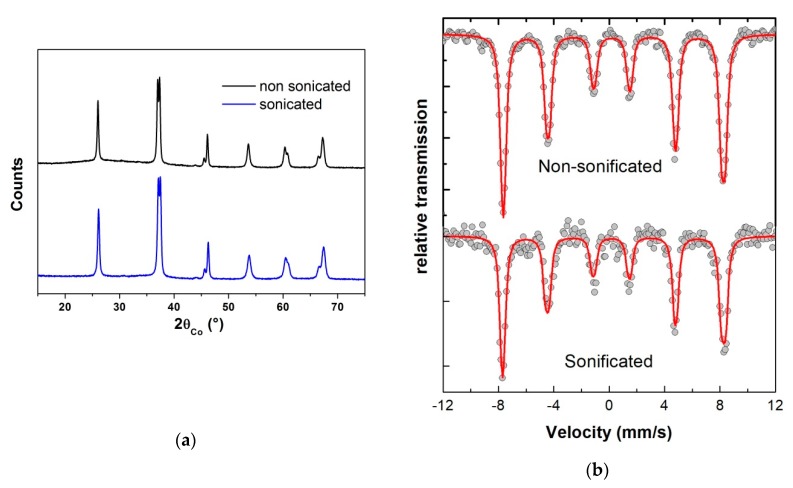
(**a**) XRD patterns and (**b**) Room temperature Mössbauer spectra of the powders obtained after heat treatment at 500 °C for 2 h of the precursors synthesized with or without sonication of the nitrates in polyol solution before heating under reflux at 200 °C for 2 h (C = 0.06 mol·L^−1^).

**Figure 6 nanomaterials-10-00026-f006:**
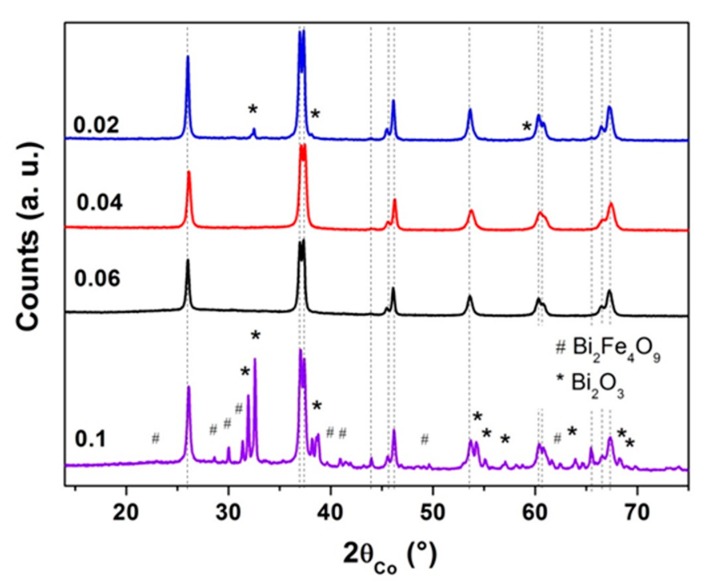
XRD patterns of the precursors prepared with concentrations of 0.02, 0.04, 0.06 and 0.1 mol·L^−1^ and heat treated at 500 °C for 2 h, illustrating the effect of the initial concentration of polyol solution on the purity of the final powders.

**Figure 7 nanomaterials-10-00026-f007:**
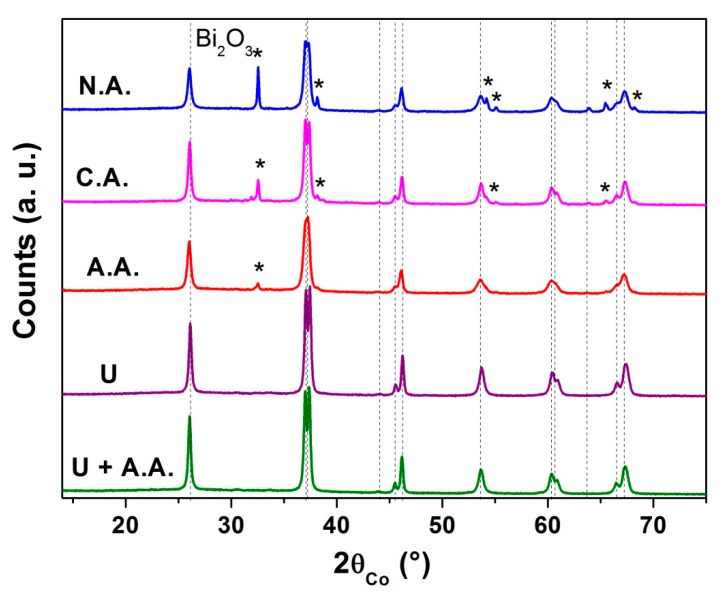
XRD patterns of the powders obtained by heat treatment at 500 °C of the precursor synthesized with the addition of acids and/or urea in order to determine the pH effect on the purity of the final powders. (N.A.: nitric acid, C.A.: citric acid, A.A.: acetic acid, U = urea).

**Figure 8 nanomaterials-10-00026-f008:**
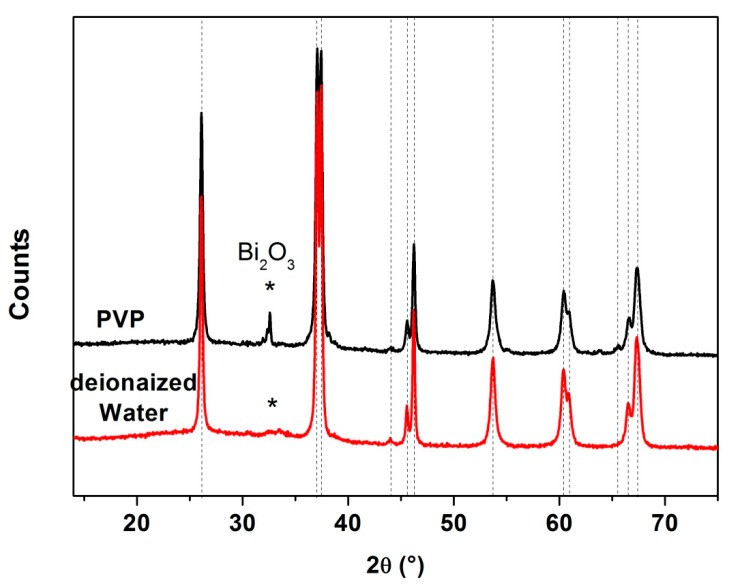
XRD patterns of powders prepared with addition of water (red line) or polyvinylpyrrolidone (PVP) (black line) to the initial solution that was heated under reflux at 200 °C for 3 h (precursor heat treated at 500 °C for 2 h).

**Figure 9 nanomaterials-10-00026-f009:**
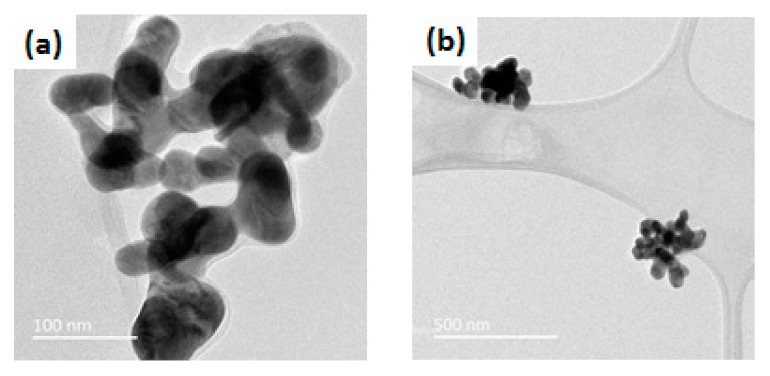
TEM images showing the precursors heat treated at 500 °C for 2 h and synthesized with (**a**) reflux for 6 h, (**b**,**c**) reflux for 3 h (**d**) reflux for 2 h, (**e**) addition of water, (**f**) addition of urea, (**g**) addition of acetic acid and urea and (**h**) addition of PVP, (C = 0.06 mol·L^−1^). The last four samples were refluxed at 200 °C for 3 h.

**Figure 10 nanomaterials-10-00026-f010:**
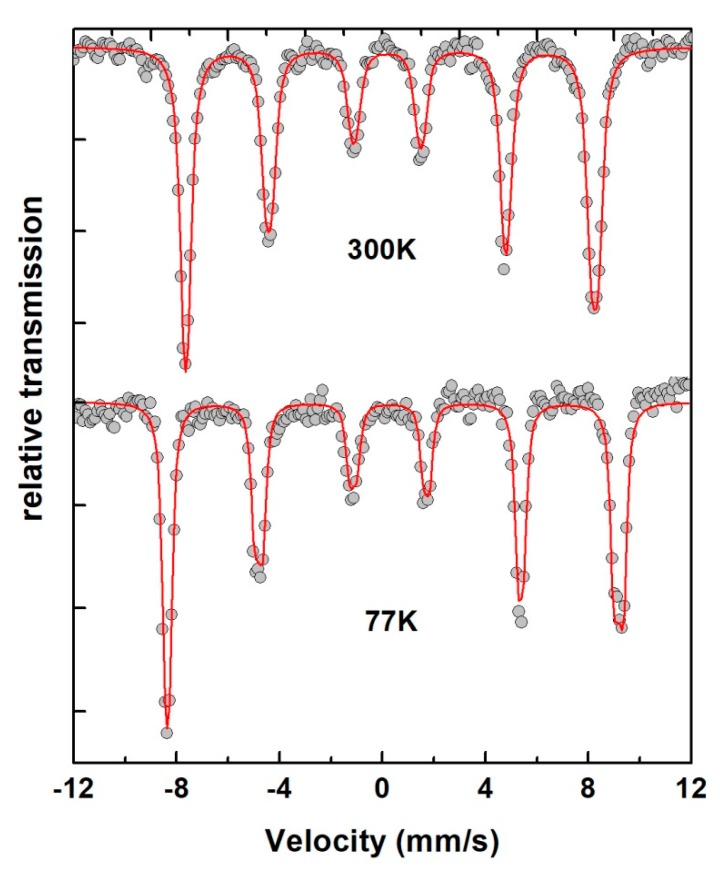
Mössbauer spectra recorded at room temperature (300 K) and at nitrogen liquid temperature (77 K) of obtained BFO nanoparticles. Symbol plot and red full lines represent experimental data and refined curves based on cycloidal model respectively.

**Figure 11 nanomaterials-10-00026-f011:**
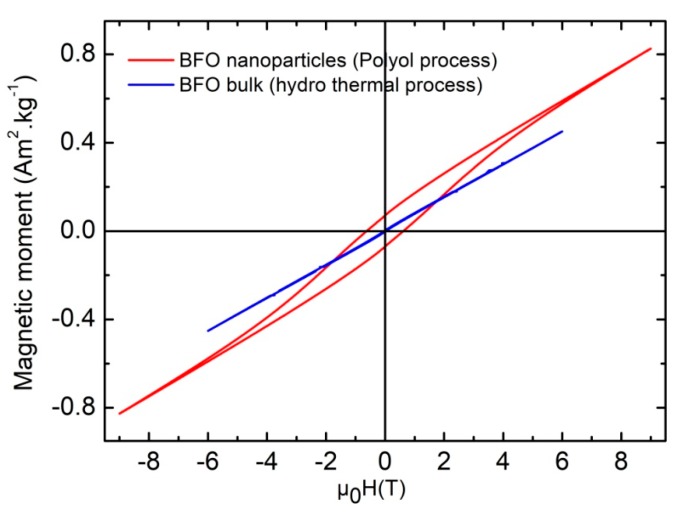
Hysteresis loops of BFO nanostructured particles synthesized by the polyol process and of BFO micrometric size particles obtained by hydrothermal process, recorded at room temperature.

**Table 1 nanomaterials-10-00026-t001:** Study of the effect of acids and urea on BFO stability and crystallite sizes (R = [acid]/[Bi + Fe] and U = [urea]/[Bi + Fe]).

Acid/Base	Quantities in 30 mL of DEG	Phases Observed by XRD	Crystallites Sizes (nm)
Nitric acid	R = 3.5	BiFeO_3_ Bi_2_O_3_ (25 wt%) *	31
Citric acid	R = 3	BiFeO_3_ Bi_2_O_3_ (14 wt%) *	40
Glacial Acetic acid	R = 1	BiFeO_3_ Bi_2_O_3_ (9 wt%) *	30
Urea (powder)	U = 2	BiFeO_3_	43
Urea + Acetic acid	U = 2 + R = 1	BiFeO_3_	43

* notice the limitation of XRD measurements to identify Bi rich phases or Bi_2_O_3_ since they represent the same cubic parameters.

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
