# Peer review of "Investigation of Polyol Process for the Synthesis of Highly Pure BiFeO3 Ovoid-Like Shape Nanostructured Powders"

_nanomaterials, 2019, doi:10.3390/nano10010026_

Round 1

Reviewer 1 Report

The scientific content of the ms is very interesting and this work thus deserves – according to my opinion – acceptance in NANOMATERIALS. I am sure that the paper will attract the intense interest of scientists working in the area of the chemistry and properties of multiferroic materials, and especially of the prototype BiFeO3 (BFO). Also, I do believe that the article will receive a respectable number of citations in the future. Salient features of this work – which support my proposal for acceptance – are: (a) The authors have systematically studied a number of synthetic parameters (nature of solvent, temperature of the reaction, reaction time, concentration, dissolution method, pH, addition of water or surfactants) on the purity and grain size of the nanostructured powders (obtained by the polyol process), and have established the optimum magnetic structure at this temperature; and (c) An impressive room temperature magnetic hysteresis was recorded, which explains well the enhancement of the ferromagnetic order compared to the antiferromagnetic bulk material. This work opens new doors for possible applications of the described low-cost BFO nanostructured powders in spintronics. The ms is well written and nicely organized. The quality of figures is high and the references list covers the topic under study satisfactorily.

Although I tried hard(!), I could not locate scientific errors and inconsistencies in the text. My minor concerns are two:

(1) “Introduction” is long and should be condensed. Much of the general information provided is well known.

(2) Part 2.1 in the “Materials and Methods” section: Perhaps I miss the point, but I can not understand the meaning of the last sentence (“It has to be pointed out …………..impurities[45]”). I would welcome an explanatory note or a modification of the sentence.

In conclusion, this work illustrates an excellent piece of research in the area of multiferroic materials and – no doubt – it deserves publication in this prestigious journal.

Author Response

Dear reviewer, thanks for your comments.

 Although I tried hard(!), I could not locate scientific errors and inconsistencies in the text. My minor concerns are two:

(1) “Introduction” is long and should be condensed. Much of the general information provided is well known.

The introduction was condensed as ask

(2) Part 2.1 in the “Materials and Methods” section: Perhaps I miss the point, but I can not understand the meaning of the last sentence (“It has to be pointed out ………….. impurities[45]”). I would welcome an explanatory note or a modification of the sentence.

If too much precursor is placed into the crucible and forms a thick layer, the combustion of the organic part will lead to an important increase of the local temperature and consequently to the formation of impurities. Moreover, the onset and homogeneity of the combustion step is enhanced by a higher contact of the powder with the air, which is favored if the layer is thin. .

We also modified the sentence.

In conclusion, this work illustrates an excellent piece of research in the area of multiferroic materials and – no doubt – it deserves publication in this prestigious journal.

Reviewer 2 Report

Authors have performed the synthesis of BiFeO changing different parameters in order to check the best synthesis conditions. In my point of view the manuscript have to be improved. I have some comments about the manuscript:

- authors said that the best synthesis temperature is 200 C. Why this temperature is the best? I think authors must explain what happen at this temperature. 

- line 204 authors said that in figure 3b authors said that they compare 2, 3 and 6 h. In the figure I only see 2 and 3 hours, 6 is missing.

- What happen after 2h that makes more crystalline the product?

- line 208: TEM in the figure 7, it is figure 8.

- line 253. To study the influence of the pH authors changed the parameters, they used 4h instead 3h. Why?

- in my point of view the TEM is not representative, it is necessary a micrography at less magnification.

- Authors performed a lot of experiments that , in my point of view, without any criteria. For instance, when they introduce HNO3, they are introducing water in the reaction... Also It has sense talking of pH in an organic medium?

- following with the pH, I think that authors must talk in numbers, I mean, pH between 1-4 we observe ... between 4 and 8... and so on so on.

- line 300: in my point of view the particles are amorfous, not ovoids.

- Line 302: authors said that adding products the particles aggregate, however, observing the TEM, I think all of them seems aggregated, maybe due to the TEM preparation...

- line 304-305: authors said that the particles have a narrow size distribution, I appreciate to have a size distribution graph.

- There is a similar paper: ceramics international 2015, 41(8), 9642-9646

Reviewer 3 Report

The authors reported the investigation of synthetic conditions of BiFeO3. The setting conditions were sound, but the experimental procedures described were not appropriate. The authors should notice the heating conditions for the formation of the precursors. If the samples are heated in air, the metal oxides are inevitably formed. Thus, all the heating procedures should be done under an inert atmosphere. I cannot understand that the authors only described the PXRD patterns after the heat treatments for 2 h. I feel the comparison of the PXRD patterns before and after the heat treatment may be important.
Therefore, I would like to reconsider the publication after the authors will address this point.

Other points
The PXRD only characterized the crystalline phases and cannot characterize the amorphous phases. The Mossbauer spectra only characterized the local environment of the iron atom. The magnetization only characterized the bulk magnetic properties. Therefore, I feel that all the measurement results for each reaction condition should be included in the text or the supporting information.

The TEM images suggest that the size of the nanoparticle of BiFeO3 is largely varied. I wonder about the homogeneity of the particles. The authors should include the EDX mapping of each TEM image.

How about the temperature dependence of the magnetic susceptibility of the nanoparticles?

Reviewer 4 Report

In this paper the authors report on the synthesis of” BiFeO3 nanopowders by polyol process. The synthesis protocol was explored and adjusted to control the purity and the grain size of the final product. The optimum parameters were carefully established and an average crystallite size of about 40 nm was obtained.” The results presented in this paper are novel and quite interesting. In principle this paper could be published in Nanomaterials but my opinion is that the manuscript is incomplete. I have observed in the manuscript that the authors have in preparation another manuscript entitled “Highly pure BiFeO3 nanoparticles synthesized by polyol process: a comparative study of the microstructure and magnetic properties with micrometric powders” where I assume (from the title) they have introduced the magnetic characterization of the BiFeO3 nanopowders. My opinion is that in order to be considered for publication in Nanomaterials the authors need to introduce in this manuscript the magnetic characterization of the here-synthesized nanopowders. Therefore, I recommend rejection of this manuscript (in the present form).  

Author Response

Dear reviewer, thanks for your comment. We do present the magnetic characterization of the best sample. The section "3.2. Magnetic properties" is entirely devoted to it. Therefore, we do not understand your recommendation. As cited in the text, we are effectively about to submit a paper devoted to the comparison of magnetic properties in micrometric and nanometric powders, in which we pay particular attention to the relationships between crystalline structure, microstructure and physical properties. This study represents a substantial work and we estimated that it should be reported in another article, this first one being more devoted to synthesis development.

Round 2

Reviewer 2 Report

In my point of view, as I said in my previous report some experiments are needed to corroborate the facts. Furthermore, I think that the authors did not answer properly all my comments. When they perform the synthesis adding nitric acid more control experiments are needed. 

Author Response

Thank you for your comments.

We want to precise that to test the effect of hydronium ions and to enhance the solubilization of iron reagent, we chose to use nitric acid because it is a stable acid (its concentration does not change with time) that can be easily eliminated by heat treatment. Nevertheless, as shown in the manuscript, the results obtained with the addition of nitric acid are not satisfactory, with the higher quantity of Bi2O3 impurity. That is why we did not try to optimize these specific conditions.

Reviewer 3 Report

I understand the synthetic condition. Then the authors should add "in air" in the text of synthesis of the BiFeO3 nanoparticles.

I have a question about precursors. What are the precursors? Are the precursors the Bi and/or Fe complexes coordinated with solvent molecules and/or additive chemicals? These are the most important information to discuss all the effects. The elemental analysis (EA) of the precursors is very informative. Therefore, the EA of the precursors, at least from the best protocol, should be included in the text or supporting information.

The authors should add the figure of the EDX mapping in the text.

Author Response

Thank you for your comments.

We answer to them in the attached document.

Reviewer 4 Report

In my opinion the magnetic characterization section is not detailed enough, 

being performed for only one sample. 

In the abstract the authors claim that "Spectacular hysteresis loop was observed by magnetic measurements despite the remaining cycloïdal magnetic structure, promoting the enhancement of the ferromagnetic order and consequently the magnetoelectric coupling compared to micrometric size powders" but they provide a very small section concerning the magnetic characterization. 

The paper is well written and it is interesting but I still believe that it can not be published in this form. Therefore I recommend reject. 

Author Response

Thank you for your comments.

We want to precise that we chose to present only one sample as this manuscript is essentially devoted to the new synthesis of BFO by the polyol process, this sample being synthesized with the best conditions. The main interest of the magnetic part is that it clearly evidences that this synthesis is efficient to promote the magnetic properties of BFO as an hysteresis loop is for the first time observed with the decrease of the particle size.